# Velocities of transmission eigenchannels and diffusion

**Azriel Z. Genack** [1,2] ✉, **Yiming Huang**[1,2,3], **Asher Maor**[1,2,4] **& Zhou Shi**[1,2,5]

The diffusion model is used to calculate both the time-averaged flow of particles in stochastic media and the propagation of waves averaged over ensembles of disordered static configurations. For classical waves exciting static disordered samples, such as a layer of paint or a tissue sample, the flux transmitted through the sample may be dramatically enhanced or suppressed relative to predictions of diffusion theory when the sample is excited by a waveform corresponding to a transmission eigenchannel. Even so, it is widely assumed that the velocity of waves is irretrievably randomized in scattering media. Here we demonstrate in microwave measurements and numerical simulations that the statistics of velocity of different transmission eigenchannels are distinct and remains so on all length scales and are identical on the incident and output surfaces. The interplay between eigenchannel velocities and transmission eigenvalues determines the energy density within the medium, the diffusion coefficient, and the dynamics of propagation. The diffusion coefficient and all scattering parameters, including the scattering mean free path, oscillate with the width of the sample as the number and shape of the propagating channels in the medium change.

The diffusion equation describes the flow of particles, waves and energy from neutrons, electrical charge, molecules and microscopic particles to light, sound, and heat[1–4]. The diffusion model begins with the assumption that scattering is local—the velocity, **v**, is randomized within a distance of the transport mean free path, $\ell$, and is determined solely by scattering within the medium and not by its overall dimensions[1–4]. The average over time of the flux within the medium is determined by Fick's first law, **j**= - D∇$u$, where **j** is the current density, $D$ is the diffusion coefficient, and $u$ is the average particle concentration or energy density[2,3]. For particles in $d$ dimensions, the Boltzmann diffusion coefficient for particles is $D_B = \frac{1}{d} v \ell$[1–4]. The diffusing quantity drops towards open boundaries and extrapolates to zero at a distance $z_b$ beyond the sample, which is proportional to $\ell$[2,5–9].

The diffusion model can also be applied to the average of propagation over random configurations in mesoscopic media, in which multiply scattered waves are temporally coherent throughout the medium[10–16]. The interference of classical and quantum mechanical waves in mesoscopic samples produces a stable speckle pattern of energy or particle density. The spatial field distribution has a correlation length of half the wavelength, $\lambda/2$[17,18], and provides a fingerprint of the wave interaction with the material. Averaging such speckle patterns over an ensemble of random sample realizations yields a smooth profile of particle or energy density[16]. In the limit in which the probability that randomly scattered wave trajectories with width $\lambda/2$, known as Feynman paths, loop back upon a typical coherence length along the trajectory tends to zero[11,12,16], the average profile in space[2,6–9] and time[19–21] is a solution of the diffusion equation[2] with boundary conditions given in terms of $z_b$[2,6–9].

As the scattering strength of the medium and the confinement of the wave increase and the dimensionality of the sample decreases, the transmission and the diffusion coefficient are increasingly suppressed by the interference of waves crossing back upon themselves within the medium. When the probability that a Feynman path will cross a typical coherence length along the path approaches unity, the wave becomes

[1]Department of Physics, Queens College of the City University of New York, Flushing, NY 11367, USA. [2]Physics Program, The Graduate Center of the City University of New York, New York, NY 10016, USA. [3]Jinhua No.1 High School, Zhejiang 321000, China. [4]Kent Optronics Inc., Hopewell Junction, New York, NY 12533, USA. [5]OFS Labs, 19 School House Road, Somerset, NJ 08873, USA. ✉e-mail: agenack@qc.cuny.edu

localized[11]. Transport is then suppressed relative to predictions of diffusion theory and quasi-normal modes of the medium become exponentially localized instead of being extended over the entire sample[22–24]. Propagation in multiply scattering samples is diffusive for samples with lengths, $L$, for which $l < L < \xi$, where $\xi$ is the localization length.

The limits of diffusive propagation may also be given in terms of the ensemble average of the dimensional conductance, $g$, which is equivalent to the average of the classical transmittance, $g = \langle T \rangle$[25,26]. The dimensionless conductance is the average electronic conductance in units of the quantum of conductance, $\frac{e^2}{h}$, while the transmittance is the sum over all pairs of flux transmission coefficients between the $N$ incident and outgoing channels of the sample, $T = \sum_{a,b}^{N} |t_{ba}|^2 = \langle \mathrm{Tr}(tt^{\dagger}) \rangle$[10,27,28]. Here, $t$ is the transmission matrix (TM) with elements $t_{ba}$. The TM is most often applied to the quasi-1D wire or waveguide geometry with constant cross section and reflecting sides. A natural choice for the channels is the set of the $N$ propagating modes of the empty waveguide. The eigenvalues of $tt^{\dagger}$ are the transmission eigenvalues, $\tau_n$, so that $T = \sum_{n=1}^{N} \tau_n$, with $\tau_n$ decreasing for increasing $n$. For $g > 1$, waves in multiply scattering media are diffusive with $g$ open transmission eigenchannels (TEs) with $\tau_n > \frac{1}{e}$[10,29], while for $g < 1$, transmission is small in all channels and waves are localized. Dorokhov showed that each of the $N$ TEs of a conducting wire scales differently with its own localization length[10]. Waves propagate diffusively for $N \gtrsim g \gtrsim 1$.

The TEs are the singular vectors found in the singular value decomposition (SVD) of the TM, $t = \mathcal{U}\Lambda\mathcal{V}^{\dagger}$[10,26,29–33]. Here $V$ and $U$ are unitary matrices whose columns are the singular vectors on the input and output of the sample, respectively, and $\Lambda$ is a diagonal matrix whose elements are the singular values, $\lambda_n = \sqrt{\tau_n}$. The amplitudes of the $m^{\mathrm{th}}$ channel in the $n^{\mathrm{th}}$ TE

on the input and output surfaces are $v_{nm}$ and $u_{nm}$, respectively. Here, the channels will be taken to be the waveguide modes.

Aside from the suppression of transport due to Anderson localization[22,34] and weak-localization precursors[11–16], dramatic deviations from diffusion theory arise in mesoscopic samples with $g > 1$ due to global correlation which produces strong variation of transmission in different TEs[10,29,31,32,35,36]. Transmission may be perfect[10,29,37–40] or vanish[41,42], and the energy within the sample may be greatly enhanced or suppressed relative to the diffusive solution[43–45] when the sample is excited by a TE. This makes it possible to control the transmission of classical waves[33,37–39,41,46–49].

The enhancement of transmission in highly transmitting TEs may be exploited, for example, to reduce the power required for cellular communication[39], while the enhancement of energy within a medium holds promise for medical imaging and intervention[33,49]. On the other hand, the suppression of transmission may enable high dynamic range switching and extreme sensitivity to sample deformation[33,41,46,48–50]. In random slabs thinner than the transport mean free path, $L < \ell$, the grain sizes of optical speckle patterns of different TEs differ[51]. This provides an approach towards engineering speckle correlation in thin samples for improved resolution for structured illumination microscopy[52].

In this study, we show that the velocity distributions of TEs on the input and output surfaces of multiply scattering media are not randomized by multiple scattering. We focus on the longitudinal components of the transmission eigenchannel velocities (EVs), $v_n$, which are the weighted averages over the angular distribution of the velocity component of the wave normal to the sample surface for different TEs. In the waveguide geometry, this may be computed as the weighted average over the distributions of group velocities of waveguide modes. The $v_n$ asymptotically approach different values as the sample length increases. As a result, the different TEs have different speckle patterns on the sample's surfaces. The interplay between the $\tau_n$ and $v_n$ yields the energy density on the open surfaces

of the sample, as well as $D$ and $z_b$. These parameters, as well as the scattering mean free path, $\ell_s$, in which spatial coherence is lost, and $g$ vary with the width of the sample as the number and shape of the transverse propagating modes change. We describe the impact of transverse boundaries on the diffusion coefficient and the Thouless conductance.

## Results
### Measurement of eigenchannel velocities
Microwave measurements of spectra of the in- and out-of-phase components of field transmission coefficients, are described in Methods and in Supplementary Fig. 1a. Spectra are obtained for two perpendicular orientations of wire antennas on the input and output of the sample on a square grid of points with use of a vector network analyser, as shown in Supplementary Fig. 1b[47]. The sample is composed of randomly positioned dielectric elements contained in a copper tube. Over the frequency range of the experiment of 14.70–14.94 GHz, the number of propagating modes supported by the waveguide changes from $N = 61,62$ to $63,64$. The wave is diffusive with $g \sim 6$. Experimental details are given in Methods.

A superposition of waveguide modes is fit to the spatial distributions of the field at points on the grid at the incident and output surfaces for each of the four antenna orientations. The TM at each frequency is then expressed in terms of the waveguide modes. The TEs on the surfaces of the sample are then obtained from the SVD of the TM. This yields continuous profiles of intensity and phase at each frequency and polarization of the source and detector, such as the intensity speckle patterns in transmission for $n = 1$ and $n = 50$, shown in Fig. 1a, b, and the corresponding phase patterns shown in Fig. 1c, d. There are fewer speckle spots for $n = 1$ than for $n = 50$.

The number of speckle spots in the intensity pattern for a single polarization on the output surface is proportional to the number of phase singularities at which the intensity vanishes and the phase changes by $2\pi$ in a loop around a singularity[53,54]. The average number of phase singularities in polarized speckle patterns on the output surface of the TEs is plotted in Fig. 1e and seen to increase with $n$. This reflects the larger range of values of the transverse k-vectors for TEs with higher $n$, and corresponds the smaller longitudinal k-vectors and smaller EVs. This is seen in the plots of the weights of waveguide modes in the TEs at the sample output, $|u_{nml}|^2$, for $n = 1$ and $50$ in Fig. 1f. The waveguide modes are indexed with $m$ increasing as the group velocity falls.

The EVs of the incident and transmitted waves for the $n^{\mathrm{th}}$ TE, $v_{n,\mathrm{i}}$ and $v_{n,\mathrm{t}}$, respectively, are given by $v_{n,\mathrm{i}} = \sum_{m=1}^{N} |v_{nm}|^2 v_{wm}$ and $v_{n,\mathrm{t}} = \sum_{m=1}^{N} |u_{nm}|^2 v_{wm}$, where $v_{wm}$ is the group velocity of the $m^{\mathrm{th}}$ waveguide mode. The average EVs for TEs on the input and output surfaces fall with $n$, as seen in Fig. 1g. This reflects the increasing contributions of waveguide modes with smaller axial velocities as $n$ increases. The small differences between the plots in Fig. 1g, h are consistent with the source antenna being more nearly perpendicular to the axis of the waveguide than the detection antenna. Such differences are absent in the numerical simulations discussed below. Numerical simulations facilitate the study of the scaling of EVs and other propagation parameters.

### Simulations of eigenchannel velocities
We carry out recursive Green's function simulations[55,56] of electromagnetic propagation polarized perpendicular to random 2D samples such as shown schematically in Supplementary Fig. 2. The samples of width $W$ and length $L$ are composed of square cells with sides of length $a = \lambda_0/2\pi$, where $\lambda_0$ is the free-space wavelength. The dielectric constant in each cell, $\varepsilon$, is drawn randomly from a rectangular distribution $[1 - \Delta\varepsilon, 1 + \Delta\varepsilon]$ with $\Delta\varepsilon = 0.3$. The dielectric constant is uniform in the direction perpendicular to the plane of the sample. The recursive Green's function method is discussed in Supplementary Note 1.

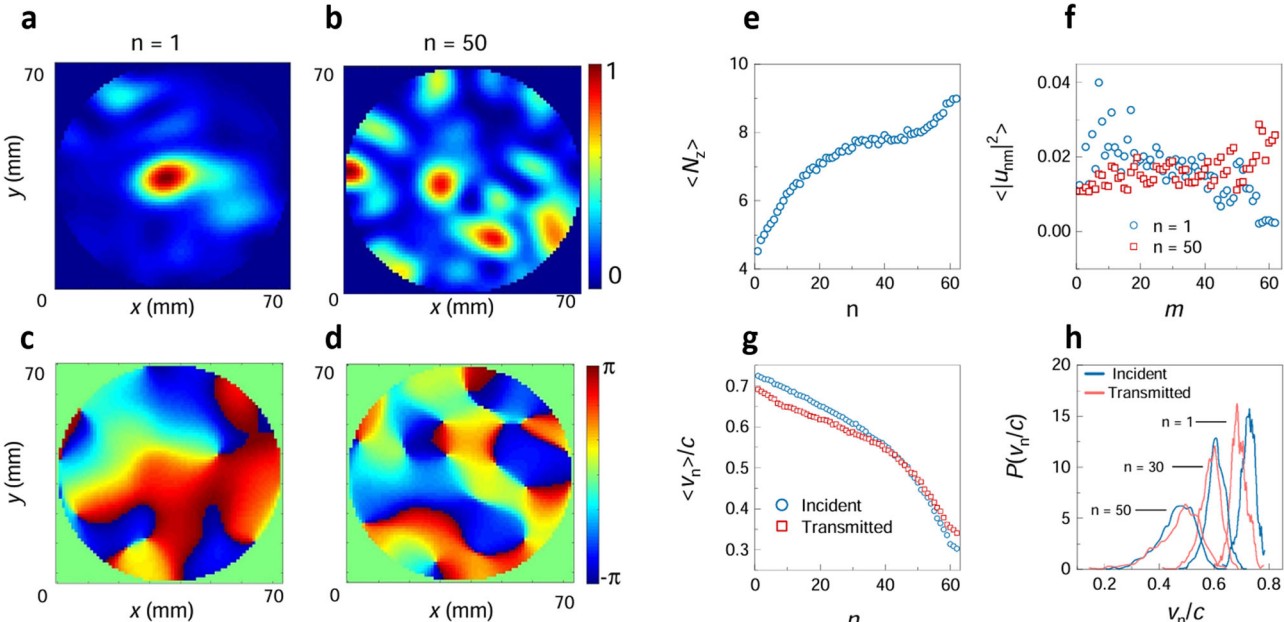

**Fig. 1 | Measurements of microwave speckle and eigenchannel velocities.**
**a**, **b** The intensity and (**c**, **d**) the phase patterns of the 1[st] and 50[th] TEs at a single frequency for a single polarization of the incident and output field in a single configuration. **e** The average number of phase singularities in polarized spectra. The intensity vanishes at a phase singularity. **f** The weight of waveguide modes in the 1[st] and 50[th] TE. **g** The EVs of the incident and transmitted waves vs. eigenchannel index, *n*. **h** The probability distribution functions (PDFs) of EVs for the incident and transmitted TE for *n* = 1,30,50. The differences between the input and output distributions in (**g**) and (**h**), are consistent with a small difference in orientation of the source and receiver antennas relative to the waveguide axis.

The scaling of the transmission eigenvalues and the EVs for $N = 8$ are shown in Fig. 2a and b, respectively. Whereas the $\tau_n$ fall exponentially beyond the localization length of each of the TEs and approach unity as $L \to 0$, the EVs of the transmitted wave $v_{n,t}$ saturate at distinct values as the sample length increases. For samples with large $N$, $v_{n,t}$ approaches its asymptotic value at lengths shorter than the localization length of approximately $N\ell_s$, as can be seen clearly in a sample with $N = 64$ in Supplementary Fig. 3 and Supplementary Note 2. This indicates that the different values of the EVs is a mesoscopic phenomenon unrelated to Anderson localization. The decrease in transmission for TEs with smaller EVs is consistent with the decrease in transmission with angle of incidence of an optical beam illuminating a random slab[9], as discussed in Supplementary Note 3.

The inverses of the EVs obey a sum rule because of their relationship to the average density of states (DOS) per unit angular frequency and length, which is uniform along the sample, $\rho_{\omega,L}(z) = \rho_\omega/L$. The relationship can be found by considering the sum of transmission times in all channels, given by $\tau_T = \sum_{n=1}^{N} t_n = \pi\rho_\omega$[57]. Since $\rho_\omega$ is independent of scattering strength in systems with the same average value of $\varepsilon$, it is the same as in a homogeneous sample[58]. In the present work, $\langle \varepsilon \rangle = 1$, and the transmission time for a TE in a uniform sample of unit length is $t_n = \frac{1}{v_n}$. The ensemble average local DOS (LDOS) can, thus, be expressed as, $\rho_{\omega,L} = \frac{1}{\pi}\sum_{n=1}^{N}\frac{1}{v_n} \equiv \frac{N}{\pi v_+}$. As a result, $v_+$ is independent of sample length, as seen in Fig. 2b.

The correlation of the EVs across the sample is seen in the identical PDFs of EVs with the same $n$ on the incident and output surfaces, while the statistics of different EVs for different $n$, shown differ, as seen in Fig. 2c for $n = 1$ and 8 in a sample with $g = 1.74$. In addition, the reflected TE is proportional to the complex conjugate of the incident TE, so that the PDFs of EVs in reflection are also identical, as demonstrated in Supplementary Note 4. Thus, the average values of the EVs for the incident, reflected and transmitted waves are identical,

$$v_{n,i} = v_{n,r} = v_{n,t} \equiv v_n. \tag{1}$$

As is conventional in discussions of the transmission eigenvalues, $\tau_n$, symbols for the EVs or other variables can refer either to the variable in a single configuration or the average over a random ensemble, depending on the context.

The EVs provide the link between the flux $\tau_n$ and the linear energy densities excited in TEs on the left and right boundaries of the sample, $u_n(0)$ and $u_n(L)$, respectively. For unit incident flux from the left, the linear energy densities of the incident and transmitted TEs are $u_{n,i}(0) = 1/v_{n,i}$ and $u_{n,t}(L) = \tau_n/v_{n,t}$, respectively. Since energy is conserved, the reflected flux in a transmission eigenchannel is $1-\tau_n$ and the energy density of a TE in reflection is $u_{n,r}(0) = (1 - \tau_n)/v_{n,r}$. The PDFs of $\tau_n/v_{n,i}$ and $\tau_n/v_{n,t}$ are seen in Fig. 2d to be identical, and these are also identical to the PDFs of $\tau_n/v_{n,r}$. The average energy density in a TE at the input is found in simulations to be the sum of the averages of the energy density in the incident and reflected waves, $u_n(0) = u_{n,i}(0) + u_{n,r}(0) = \frac{1}{v_{n,i}} + \frac{(1-\tau_n)}{v_{n,r}}$. The absence of interference terms between the incident and reflected waves in the average energy density at the sample input surface is shown in Supplementary Note 5 to be a consequence of the proportionality of the reflected wave and the complex conjugate of the incident wave. The average energy density excited from the left is the sum over TEs, $u(z) = \sum_{n=1}^{N} u_n(z)$. This gives

$$u(0) = \sum_{n=1}^{N} \frac{2 - \tau_n}{v_n}, \tag{2a}$$

$$u(L) = \sum_{n=1}^{N} \frac{\tau_n}{v_n}, \tag{2b}$$

on the left and right sides of a dissipationless sample.

### The diffusion coefficient
When the energy density within the sample excited from the left falls linearly, it is possible to define a diffusion coefficient via Fick's first law as the ratio of the flux and the magnitude of the gradient of the energy

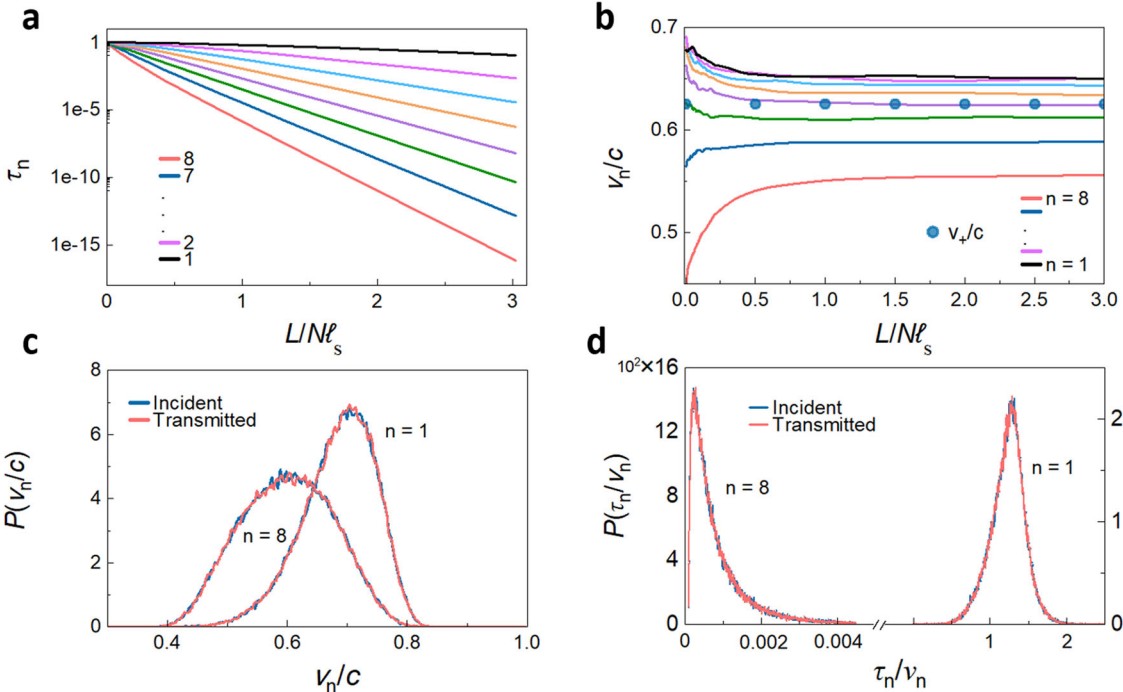

**Fig. 2 | Simulations of scaling and statistics of eigenchannel velocities. a, b** Scaling of transmission eigenvalues and EVs in a random medium with $N = 8$. **c** PDFs of EVs of the incident and transmitted waves. For the same n, the PDFs overlap. **d** PDFs of $\tau_n/v_n$ for given $n$ at the input and output boundaries also overlap.

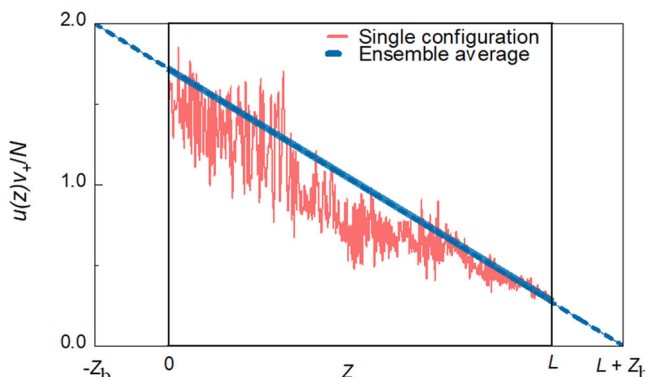

**Fig. 3 | Energy density inside the random medium.** Profiles of the normalized energy within a medium excited from the left with unit flux in all channels in a single configuration (red curve) and averaged over 2000 configurations (blue curve) for a medium with $N = 8$, $L = 600a$, and $g = 1.74$. The linear energy density is normalized by its spatial average obtained for TEs with unit incident flux. The average normalized energy density falls linearly and extrapolates to 2 and 0 at a distance $z_b$ in front of and behind the sample, respectively.

density, $D = -g/\frac{du}{dz}$, with $\frac{du}{dz} = \frac{u(L) - u(0)}{L}$, giving

$$D = \frac{gL}{u(0) - u(L)}. \qquad (3)$$

With the energy densities at the sample boundaries given in equation (2), this gives

$$D = \frac{gL}{2\left[\sum_{n=1}^{N}\frac{1}{v_n} - \sum_{n=1}^{N}\frac{\tau_n}{v_n}\right]}. \qquad (4a)$$

The diffusion coefficient may also be expressed in terms of $v_+$ and an effective transmission velocity, $v_T$, defined via the relation,

$u(L) = \sum_{n=1}^{N}\frac{\tau_n}{v_n} \equiv \frac{g}{v_T}$, as

$$D = \frac{gL}{2\left[\frac{N}{v_+} - \frac{g}{v_T}\right]}. \qquad (4b)$$

The diffusion coefficient can be defined as in Eqs. (3) and (4) even for $g = 1.74$, which is close to the crossover to Anderson localization, since $du(z)/dz$ is essentially constant throughout the sample, as seen in Fig. 3. $u(z)$ is normalized in the figure by its spatial average, $\langle u(z)\rangle_z = \frac{N}{v_+}$, which is calculated in Supplementary Note 6. The figure also shows the normalized energy density in a single configuration. The configuration average of $u(z)v_+/N$ falls linearly in the sample and extrapolates to zero a distance $z_b$ to the right of the sample and to 2 at a distance $z_b$ to the left of the sample. Since the triangles in Fig. 3 with base of $L + 2z_b$ and height 2 and with base $z_b$ and height $u(L)v_+/N$ are similar, $\frac{2}{L + 2z_b} = \frac{u(L)v_+/N}{z_b}$[59], and

$$u(L) = \frac{2Nz_b/v_+}{L + 2z_b}. \qquad (5)$$

Since $g = u(L)v_T$, this gives

$$g = \frac{2Nz_b}{L + 2z_b}\frac{v_T}{v_+}. \qquad (6)$$

Substituting Eq. (6) into Eq. (4b) gives

$$D = z_b v_T, \qquad (7)$$

as shown in Supplementary Note 7. Unlike, the Boltzmann diffusion coefficient, $D_B$, the diffusion coefficient, $D$, is expressed here in terms of the nature of the wave near the boundaries rather than in the bulk of the medium.

Comparing Eq. (7) to the classical expression for the diffusion coefficient in $d$ dimensions, $D_B = \frac{1}{d}v_E\ell$, gives, $\ell = dz_b v_T/v_E$. As seen

**a**

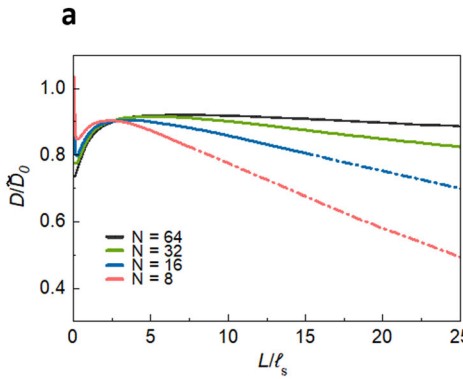

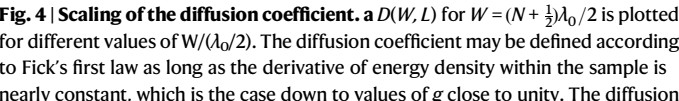

**b**

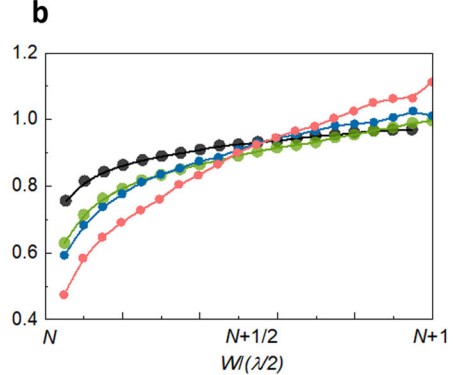

**Fig. 4 | Scaling of the diffusion coefficient. a** $D(W, L)$ for $W = (N + \frac{1}{2})\lambda_0/2$ is plotted for different values of $W/(\lambda_0/2)$. The diffusion coefficient may be defined according to Fick's first law as long as the derivative of energy density within the sample is nearly constant, which is the case down to values of $g$ close to unity. The diffusion coefficient, as given by Eq. (3) and (4), is plotted as dashed curves once $g < 1$ since $u(z)$ then no longer falls linearly. **b** The variation of the diffusion coefficient with wavelength for each $N$ is plotted for sample lengths at which the diffusion coefficient in (a) is at its maximum.

below, $D$ and its factors $z_b$ and $v_T$ vary with the dimensions of the sample, so that $\ell$ that appears in the Boltzmann diffusion coefficient is not an intensive parameter for diffusing waves, as is often assumed. In the extreme diffusive limit, however, in which $N \gg g \gg 1$, $D$ and its factors $z_b$ and $v_T$ would be expected to approach the classical particle limit, in accord with the correspondence principle.

The variation of $D$ with length and width for an ensemble with $\Delta\varepsilon = 0.3$ and $N = 8, 16, 32,$ and 64 channels is shown in Fig. 4a and b, respectively. To place the results in the context of particle diffusion, $D$ is normalized by $\widetilde{D_0} = \frac{1}{d}c\ell_s$, where $d = 2$ is the dimensionality, $c$ is the speed of the wave in a medium with $\varepsilon = 1$, and $\ell_s$ is the scattering mean free path, which is the distance in which the wave loses coherence. $\ell_s$ is determined from the identical decay times of the coherent flux for all waveguide modes[59], as discussed in Supplementary Note 8 and shown in Supplementary Fig. 4. $\widetilde{D_0}$ approximates the bare particle diffusion coefficient, $D_0 = \frac{1}{2}v_E\ell^{2,3,60}$. Here, $v_E$ is the energy transport velocity[60], which may be lower than the phase velocity in the medium as a result of resonances with scattering elements. However, since the sides of the scattering elements of the sample are considerably shorter than the wavelength, $a = \lambda_0/2\pi$, these elements are far from resonance. Scattering should therefore be nearly isotropic, with $v_E \sim c$, $\ell_s \sim \ell$, and $\widetilde{D_0} = \frac{1}{2}c\ell_s \sim D_B$.

The variation of $D$ with length for samples with width in the centre of the range for each channel, $W \sim (N + \frac{1}{2})(\lambda_0/2)$ is shown in Fig. 4a. $D(L)$ reaches a peak after several scattering mean free paths and then falls due to weak localization[12,61]. The decay is more rapid for samples with smaller $N$ since the crossover to localization at $g \sim 1$ at length $\xi \sim N\ell \sim N\ell_s$, is reached at shorter lengths. The value of $D$ found from the decay of pulsed transmission following the peak in transmission is the same as obtained in steady-state simulations, as shown in Supplementary Note 9 and Supplementary Fig. 5.

The diffusion coefficient also changes as the ratio of the sample width and wavelength is varied by changing the wavelength in a sample with the same number of square scattering elements with sides of length $\lambda/2\pi$. The variation of $D$ over the range of width of $[N, N+1]\lambda_0/2$ for the same values of $N$ and $\lambda_0$ as in Fig. 4a at values of $L/\ell_s$ at which $D/\widetilde{D_0}$ reaches its peak is shown in Fig. 4b. $D$ varies because the shapes of the propagating modes changes with sample width. The fractional variation of $D$ over the range of wavelengths for a given $N$ decreases as $N$ increases but is still appreciable for $N = 64$. We note that the crossover to a new channel differs slightly from the value $N = W/(\lambda/2)$ because of the discretization of the sample in the simulations.

The variation of $z_b$ and $v_T$ with length for samples with $W = (N + \frac{1}{2})\frac{\lambda_0}{2}$ for $N = 8, 16, 32, 64$ is shown in Figs. 5a and 5b. Since $v_T$ varies little for $L/\ell_s > 2$, the fall in $D/\widetilde{D_0}$ with $L$ after the crossover from

ballistic to diffusive propagation primarily reflects the variation of $z_b/\ell_s$. The values of $z_b$ in Fig. 5a are obtained by solving Eq. (6) to give

$$z_b = gL/2\left[N\frac{v_T}{v_+} - g\right] \qquad (8)$$

Equation (8) shows that $z_b$ depends upon the EVs as well as upon the transmission eigenvalues.

The values of $z_b/\ell_s$ in the centre of the wavelength range for a given $N$ are close for all values of $N$, as seen in Fig. 5c. For $N = 64$, $z_b/\ell_s = 0.711$. This is close to the value obtained from the solution of the Milne problem of 0.7104 for equilibrium radiative transfer near the surface of a half space due to a remote source within the medium[2,5].

The variation with sample width of $z_b/\ell_s$, $v_T/c$, $z_b$, and $\ell_s$ is shown in Fig. 5c-f for sample lengths at which $D/\widetilde{D_0}$ reaches its peak value for each value of $N$ at sample widths $W = (N + \frac{1}{2})\frac{\lambda_0}{2}$. The variation of these parameters with width for $N = 8$ and 64 and a large range of lengths is shown in Supplementary Fig. 6 and Supplementary Note 10.

Closer to the Anderson localization threshold, the diffusion model breaks down with $u(z)$ falling faster in the centre of the sample than at the boundaries, as seen in Fig. 6a, so that it is not possible to define a diffusion coefficient. For $g = 1.078$, $u(z)$ falls 10% faster at the centre than at the edges of the sample, as is seen in Supplementary Fig. 7 and discussed in Supplementary Note 11. Transport can then be described in terms of a position-dependent diffusion coefficient, $D(z) \equiv -g/\frac{du(z)}{dz}^{62,63}$, which dips in the middle of the sample because of the larger probability of trajectory crossing themselves there than near the sample boundaries.

The energy density profiles in Figs. 3 and 6a, and Supplementary Fig. 7 are the sums of the energy density of TEs, $u_n(z)$, such as the profiles shown in Fig. 6b-d, with values at the boundaries that depend upon $\tau_n/v_n$, as in Eq. (2). In addition to the larger magnitude of the slope of $u(z)$ in the centre relative to that at the boundaries of the sample as $g$ decreases, as seen in Fig. 6a and Supplementary Fig. 7, there is an inversion in the ranking of energy excited in the sample vs. eigenchannel index, $n$, in the crossover from ballistic to diffusive propagation. In translucent samples, all of the $\tau_n$ are close to unity and the energy density throughout the sample is dominated by the factor $1/v_n$, and so the energy within the sample is larger for smaller transmission eigenvalues, as seen in Fig. 6b. This trend is reversed in longer samples, as seen in Fig. 6c, d. For $L > \ell_s$, the variation of $\tau_n/v_n$ with $n$ is predominantly due to the strong variation of $\tau_n$ with $n$, as seen in Fig. 2a, and not to the weaker variation of $v_n$, as seen in Fig. 2b, and the energy excited in the sample decreases with $n$, as seen in Fig. 6c and d.

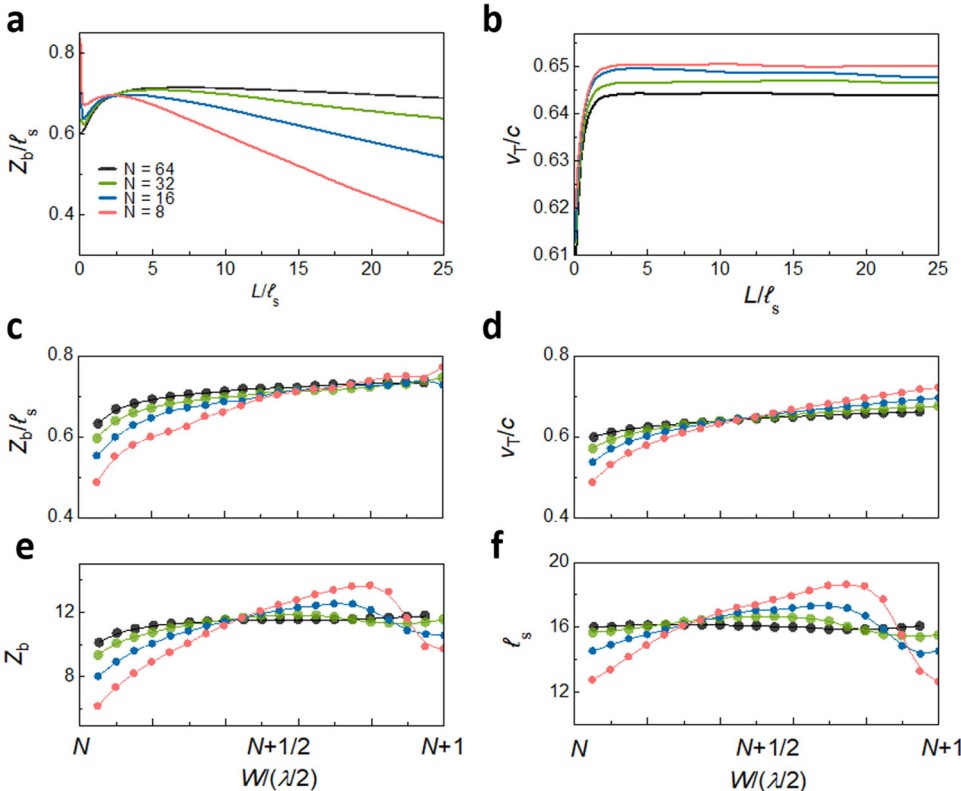

**Fig. 5 | Scaling of the factors of the diffusion coefficient. a, b** The scaling of $z_b/\ell_s$ and $v_T/c$ with length for various values of $N$. $z_b$ falls linearly with length while $v_T$ become nearly independent of $L$ after two scattering lengths. **c–f** The parameters $\frac{z_b}{\ell_s}$ (**c**), $\frac{v_T}{c}$ (**d**), $z_b$ (**e**), and $\ell_s$ (**f**) all vary with sample width. The variation of $\ell_s$ with $W$ demonstrates that the scattering process is nonlocal.

**The Thouless conductance**

The scaling theory of localization[24,64], according to which the variation of $g$ with the dimensions of the sample depends only upon $g$, is built upon the relationship between static and dynamic aspects of wave transport. The classical geometric model of the scaling of conductance in the diffusive limit, $g = \frac{A\sigma}{L\left(\frac{e^2}{\hbar}\right)}$, may be expressed in terms of dynamic parameters via the Einstein relation for the conductivity, $\sigma = e^2 D\rho_{E,A,L}$ in terms of local parameters. Here, $\rho_{E,A,L}$ is the LDOS per unit energy and volume, $\rho_E/AL$[24]. This gives $g = \frac{\hbar D\rho_E}{L^2}$. The classical wave analog of this relation is $g = \frac{2\pi D\rho_\omega}{L^2}$.

Using the results in the previous section, we obtain a relationship between $g$, $D$, and $\rho_\omega$ by substituting Eq. (7) into Eq. (6) and utilizing the relation, $\rho_\omega = \frac{NL}{\pi v_+}$, to give

$$g = \frac{2\pi D\rho_\omega}{L\left(L + 2z_b\right)}. \tag{9}$$

Equation (9) is valid as long as $u(z)$ falls linearly within the sample, which is the case even close to the localization threshold.

Following Thouless, the right-hand side of Eq. (9) may be expressed in terms of the degree of spectral overlap of the quasi-normal modes, or resonances, of the open medium, which is the ratio of the modal linewidth and the spacing between modes, $\delta = \delta\omega/\Delta\omega$ and is known as the Thouless number or the Thouless conductance, $\delta \equiv g_{Th}$. The linewidth of quasi-normal modes in a diffusive sample, $\delta\omega = \frac{\pi^2 D}{\left(L + 2z_b\right)^2}$, is equal to the decay rate of stored energy following pulsed excitation and also to the decay rate of the lowest diffusion mode of the diffusion equation[19], while the DOS is equal to the inverse of the average spacing between modes, $\rho_\omega = \frac{1}{\Delta\omega}$. Even as $D(W,L)$

is renormalized by weak localization, its value as determined from Fick's first law is still identical to that obtained from the decay rate of energy following pulsed excitation, as seen in Supplementary Fig. 5. Equation (9) can thus be expressed as $g = \frac{2(L+2z_b)}{\pi L}\frac{\delta\omega}{\Delta\omega} \equiv \frac{2(L+2z_b)}{\pi L}\delta$.

The Thouless conductance may be compared to the degree of crossing of wave trajectories, which is the ratio of the average time for waves to traverse the sample, $\tau_{Th}$, to the time to visit each coherence volume in the sample, $\tau_T$, $\frac{\tau_{Th}}{\tau_T}$. Since $\tau_T = \pi\rho_\omega$ and, $\tau_{Th} = \left(L + 2z_b\right)^2/D$[21], and $\Delta\omega$ and $\delta\omega$ are as given above, this gives, $\delta = \pi\frac{\tau_{Th}}{\tau_T}$. Equation (9) can be written as, $g = \frac{2(L+2z_b)}{L}\frac{\tau_T}{\tau_{Th}}$. Thus, whether because of low modal overlap, $\delta$, or a large fraction of coherence lengths along the trajectory that are crossed by a trajectory, $\frac{\tau_{Th}}{\tau_T}$, $g$ is suppressed below the predictions of diffusion theory as $g$ falls towards and below unity[11,16,24,64–66]. Mesoscopic fluctuations are then greatly enhanced over the level predicted by Gaussian field statistics[13,16,65,67,68], transmission spectra are sharply peaked[69–71], and energy density falls exponentially from the point at which it is injected[72–74]. Since $1/g$ expresses the degree of departure from diffusion theory, the longitudinal scaling of the conductance should depend only upon $g$ itself[24,64]. However, the oscillations in scattering parameters with sample width show that knowledge of $g$ alone is not sufficient to determine the scaling with transverse dimensions. An important question is whether the scaling of conductance may be fully determined once a second dimensionless parameter, the number of channels, is added to $g$.

**Discussion**

The absence of randomization of the velocity of transmission eigenchannels in mesoscopic samples ushers in the EVs, $v_n$, as a new set of parameters of the TM alongside the transmission eigenvalues, $\tau_n$. The average values and statistics of different EVs are different, but they are identical on the input and output surfaces of the sample for the same

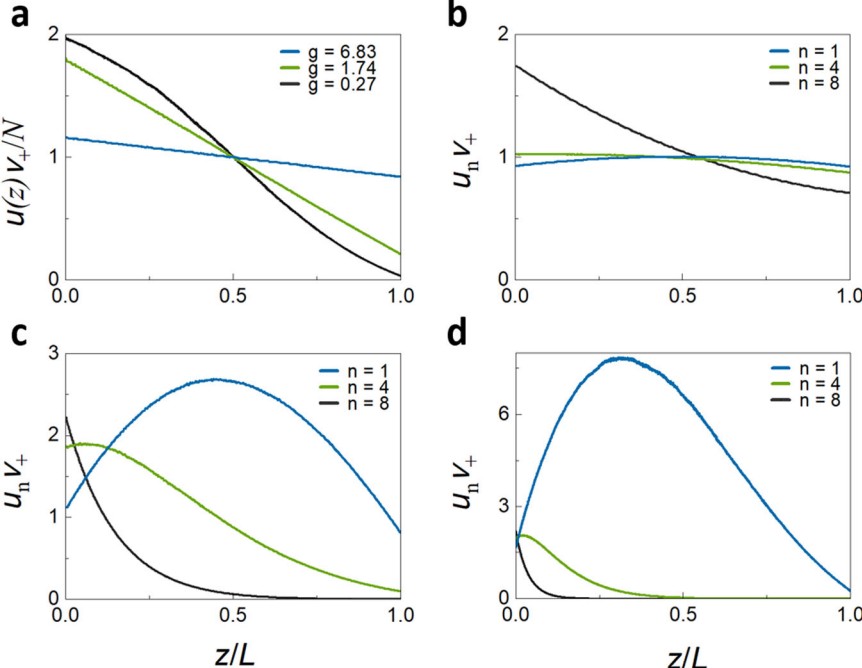

**Fig. 6 | Profiles of excitation inside the medium. a** The sum over all TEs of the average energy density excited within the medium from the left for ballistic, diffusive, and localized waves. The energy density falls linearly throughout the sample for ballistic and diffusive waves but falls more rapidly near the centre for localized waves. **b**–**d** The energy excited in TEs increases with $n$ for ballistic waves but decreases with $n$ for diffusive and localized waves.

TE. This makes it possible to utilize the TM to explore dynamic as well as steady-state propagation,

The energy densities of different TEs on the output surface may be expressed using the sets of $\tau_n$ and $v_n$ since $u_n(L) = \tau_n/v_n$. This allows the gradient of energy density within the sample and the diffusion coefficient to be found. Transport in steady-state and in the time domain can thus be described via Fick's first and second laws, respectively, in terms of the TM. Values of the diffusion coefficient found from simulations of steady-state and pulsed transmission are the same.

We have found that the diffusion coefficient, $D$, the boundary extrapolation length, $z_b$, and the effective longitudinal velocity in transmission, $v_T$, may vary appreciably over the range of wavelength or sample width in which the number of channels exciting the sample increase by unity. Since all these quantities dip near the crossover to a new channel as the sample width or wavelength is changed, even in the diffusive regime, these parameters are global rather than local and their variation with the transverse dimensions of the sample is only tangentially related to wave localization. The degree of modulation of these parameters with $W/(\lambda/2)$ falls as the size of the sample increases in harmony with the correspondence principle.

The nonlocality of propagation is seen in the expression for the diffusion coefficient as a product of two factors which reflect propagation at the sample boundaries rather than in its interior, $D = z_b v_T$. This expression for $D$ differs from the Boltzmann diffusion coefficient, $D_B = \frac{1}{d} v \ell$, expressed in terms of local parameters, which are generally assumed to be independent of the sample's dimensions.

We have also found in simulations that even the scattering mean free path, $\ell_s$, which might be expected to represent local scattering, dips near the crossover to a new channel. The scattering elements in the simulations are small relative to the wavelength, so that $\ell \sim \ell_s$. The ratio $z_b/\ell_s \sim z_b/\ell$, which has been assumed to have a constant value of 0.7104 obtained in the solution of the Milne problem in an unbounded sample[2], also varies with sample width. When the sample width is close to the centre of the range of widths for a given value of $N$, however, this ratio is close to the particle diffusion value found in the solution of the Milne problem.

Systematic variations of the conductance and transmittance as the number of propagating channels changes also arise in ballistic and diffusive samples. Stepwise increases in the conductance are measured in ballistic heterojunctions as the width is changed by a gate voltage[75] and also in the optical transmittance of diffuse light through an adjustable aperture[76]. Simulations of conductance in diffusive quasi-1D samples were carried for the Anderson model of a tight-binding Hamiltonian with diagonal disorder. Instead of steps, dips were found in the conductance near the threshold to each new channel[77,78]. The origin of the dips will be shown in future work to emerge from the correlation within and between the sets of $\tau_n$ and $v_n$.

The modulation of scattering with the number of propagation channels is reminiscent of the Wigner cusp in the nuclear scattering cross section that arises when new scattering channels open up as the energy of incident particles increases[79–81]. The enhanced variation of scattering around the crossover to a new channel can be the basis for enhanced sensitivity of classical waves to variations in the sample dimensions[82]. Present research is exploring the extreme sensitivity of EVs to changes in the sample dimensions at the crossover to a new channel. Immediate questions that emerge from this work are the relationships between the EVs and the transmission eigenvalues and the profiles of EVs within the sample.

## Methods
### Microwave measurements
In- and out-of-phase spectra of field transmission coefficients between source and receiver antennas on opposite sides of the sample are shown in Supplementary Fig. 1a. Such spectra are obtained for each of four pairs of polarizations between the input and output surfaces for each pair of locations of the 4-mm-long source and receiving antennas on a square grid with 9-mm spacing on the sample's surfaces. A schematic of the experimental setup is shown in Supplementary Fig. 1b. A copper sample tube of inner-diameter 7.3 cm is filled to a

length of 23 cm with 0.95-cm-diameter alumina spheres of refractive index 3.14 embedded in the centres of Styrofoam shells to yield an alumina volume fraction of 0.07. An ensemble of 23 random sample configurations is created by briefly rotating the tube about its axis and vibrating the sample. The sample is vibrated to allow the sample to settle so that it is static over the 40-hour period required to measure the spectrum of the TM. The antennas are formed by stripping the outer conductor and bending the central conductor of a microwave cable by approximately 90°. The bent central conductors of the source and detector are brought close to the input and output surfaces and lie parallel to the respective surfaces.

The electric fields launched into and emerging from the sample are polarized along the direction of the respective wire antennas. The fluxes on the incident and output surfaces are found by expressing the field amplitudes of the incoming and outgoing waves as superpositions of waveguide modes. The propagating modes of the empty waveguide are either transverse electric or transverse magnetic modes with the same group velocity[83]. A single crossover for waveguide modes from $N = 61,62$ to $N = 63,64$ occurs at 14.8317 GHz over the frequency range of the experiment.

## Data availability
The datasets generated during and/or analysed during the current study are available from figshare at https://doi.org/10.6084/m9.figshare.25352956.

## Code availability
The simulation codes used in the current study are available from the corresponding author upon request.

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

## Acknowledgements
We thank Krishna Joshi for plotting Fig. 1a–d and for discussions and thank Israel Kurtz for discussions. This work is supported by the National Science Foundation (US) under EAGER Award No. 2022629 and NSF-BSF Award No. 2211646 (AZG).

## Author contributions
A.Z.G. directed the project, calculated propagation parameters and wrote the manuscript with input from all authors. Z.S. carried out the measurements. Y.H. analysed the experimental data, carried out and analysed numerical simulations, and calculated the relationship between the incident and reflected TEs. A.M. carried out and analysed numerical simulations.

## Competing interests
The authors declare no competing interests.
