## [Peer Review File · Nature Communications]

Velocities of transmission eigenchannels and diffusionREVIEWER COMMENTS

Reviewer #1 (Remarks to the Author):

The authors investigate, both experimentally and numerically, the propagation of waves in disordered quasi-1D waveguides. In particular, they investigate how individual transmission channels influence transport and to what extent a description in terms of a diffusion process remains valid. While it is not the first time that the authors investigate transport and transmission channels in this configuration, there are some very interesting new insights that come out of this study. I find particularly interesting, for instance, the effect of individual channels over various length scales and how the average properties like the diffusion coefficient depend on sample width.

Having said that, I think that the paper is not particularly well-written. The introduction, for instance, is very long and boring in several places. In particular, it is not very well accessible for non experts, and it even contains several grammatical imprecisions and passages that are unclear. The paper as a whole is written in a very technical way which makes it difficult for non-experts to get excited about the results. Moreover, a discussion about their impact is missing which makes it difficult to understand how the insights obtained in this paper are relevant in other contexts. (While I believe they are, this is not explained well by the authors.)

So concluding, the paper reports on very interesting work in an active field of research. The results should have consequences in various contexts given the interdisciplinary aspects of this work, but this is, I think, not well enough explained by the authors. The presentation of the work should be improved to make it suitable for the broad readership of Nature Communications.

Reviewer #2 (Remarks to the Author):

The authors study the velocities of the eigenchannels of a strongly scattering sample in a waveguide geometry. They present measurements as well as numerical simulations.

The manuscript is interesting and represents a clear advance in understanding of the phenomenon of eigenchannels and their relation to transport physics. However, the current manuscript has a number of flaws that lead me to advise against publication in its current form.

Overall, the text is quite poorly written both on the level of structure and grammar. I give a number of examples, this is not an exhaustive list:

- Lines 12-16: Many of these examples concern incoherent or particle diffusion, in which interference phenomena such as eigenchannels play no role at all. The findings of this study are totally irrelevant to those phenomena.
- Nowhere is a clear definition of the velocity of an eigenchannel to be found. One has to assume that what is meant is the properly weighted average of the group velocity.
- L39: The sentence is really unnecessarily hard to read.
- L49 in the case of particle diffusion, no speckle pattern occurs. In the case of quantum diffusion there can be a speckle of probability density but not of particle density.
- L74 here it is unclear what the t_n represent. The authors use τ for times and t for transmission coefficients, except sometimes they use t for times and τ for transmission coefficients. Please clarify by clearly defining each symbol, and if possible use the same symbol for the same quantity.
- L80-81 Ultracold atoms.. this line seems out of place.
- L92 define τ_n
- L106 v_n is not defined precisely
- L135 how are the eigenvalues sorted? High to low or low to high?
- L176 repetition of the formula?
- L177 This seems to be true only in an ensemble average sense
- L178 LDOS should be a strongly fluctuating function (unless one means after ensemble averaging)
- L230 scattering length – do the authors mean scattering mean free path?
- 248 Unlike, ... The sentence makes no sense
- L302 Falls 10% higher
- L368 Multiple scattering

On pages 19 and further the influence on the width of the conductance, and especially the nonmonotonic behavior upon opening a new channel is discussed. This effect has been observed in calculations and numerical simulations, see e.g. <http://iopscience.iop.org/0295-5075/24/4/006> and Garcia-Mochales, P., et al. "Conductance in disordered nanowires: Forward and backscattering." *Physical Review B* 53.15 (1996): 10268.

It is well known that in the so-called semi-ballistic or quasi-ballistic regime, where some of the sample dimensions are not large with respect to the scattering mean free path, a remnant of the resonance at opening of a new channel remains. Since the dimensions of the system in these directions is not much larger than the mean free path, the conditions for applying the scaling theory in these directions are not fulfilled and the discussion on L368-373 makes no sense.

L395 Please give concrete examples of how the results of the present work relate to specific applications in medical imaging and interventions, telecommunication, and sensing. Or put a less definitive statement.

Response to the reviewers:

In the name of all the authors, I would like to thank the reviewers for their careful evaluation of our paper. Both reviewers suggested that the presentation should be simplified and that the impact of the work be stated more plainly. We thank Reviewer #2 for the careful and extensive point-by-point discussion of lapses in the text and for pointing out a stream of research that I was not aware of at the time the paper was submitted regarding the modulation of the conductance with sample width. I have revised the manuscript of the main text to address each of the points raised by the reviewers and combed the text for any other instances in which the text could be simplified. The changes made to the article are mostly in the Abstract, Introduction, and Discussion. The figures have not been changed, and the only changes made in the Supplementary Information are the replacement of “scattering length” with the “scattering mean free path,” and “scattering time” with “scattering mean free time,” as suggested by Reviewer #2.

I believe I have clarified all the issues raised by the reviewers. I have strived to make the introduction and concluding discussion less technical. I have eliminated the discussion in the Introduction of the crossing of wave trajectories and of the diffusive limit, in which the probability of crossing tends to zero. I have also highlighted the novelty and potential impact of this work. In particular, the revised manuscript emphasizes the enhanced power of the transmission matrix (TM) afforded by the new set of parameters of the TM, with the eigenchannel velocities (EVs), standing alongside the transmission eigenvalues. This is now stressed in the Abstract and Discussion.

In the Abstract, we now have:

“The interplay between the sets of eigenchannel velocities and transmission eigenvalues determines the the energy density within the medium, the diffusion coefficient, and the dynamics of propagation.”

The first sentence of the Discussion is now:

“The absence of randomization of the velocity of transmission eigenchannels in mesoscopic samples ushers in the EVs, v_n , as a new set of parameters of the TM alongside the transmission eigenvalues, τ_n .”

Reviewer #1

The reviewer writes “Moreover, a discussion about their impact is missing which makes it difficult to understand how the insights obtained in this paper are relevant in other contexts. (While I believe they are, this is not explained well by the authors.)”

---We have rewritten the Discussion section to address this point. Below, I sketch the flow of ideas in the Discussion that emphasize the insights obtained from the paper that are “relevant in other contexts,” but also insights that are important for the quantitative study of random media.

Sketch of Discussion section:

-velocities of TEs not randomized

-so have a new set parameters of the TM, the v_n , alongside the transmission eigenvalues, τ_n

$-u_n(L) = \tau_n/v_n$, so we can obtain the energy density at the boundaries, the gradient of energy density, and the diffusion coefficient D .

-The diffusion coefficient D obtained in this way gives the same result in simulations of steady-state and pulsed transmission

- D varies over the range of wavelength or sample width in which the number of channels exciting the sample increase by unity

-this variation is not due to localization

-the modulation falls as the size of the sample increases

-the nonlocality of propagation is seen in the expression for D as a product of boundary parameters

-in contrast, the Boltzmann diffusion coefficient D_B is expressed in terms of local variables of the bulk

-the scattering mean free path also varies with the transverse dimensions of the sample and it does not have a fixed ratio with the extrapolation length

-but at the center of the range between crossovers, this ratio is close to the classical Milne result

-the conductance has also been found in simulations to be modulated as new channels enter

-we allude to the fact that work carried out after this paper was submitted (with different coauthors) provides the source of the modulation in conductance in terms of the τ_n and the v_n and the interaction between them. This shows the significance of the present work but is beyond the scope of this paper.

-the modulation of scattering with the number of propagation channels is reminiscent of the Wigner cusp anomaly in nuclear scattering. (in work carried out after this work was submitted for publication, we are able to explain both the dips in conductance and the peaks in the DOS at the crossover to a new channel in terms of the τ_n and the v_n .)

-perhaps the greatest theoretical challenge raised by this work is to find the relationship between the τ_n and the v_n (The simplest such relationship would be for v_n to be a function of τ_n)

-since the energy density on the sample's surfaces can be obtained with use of the EVs, the findings in this paper should help in the search for an expression giving the energy profile within the surface.

This outline of the Discussion touches both on broad topics that will be impacted by this work and new insights and results on waves in random media.

I should say a few words about research in our group that was spurred by the present article. I cannot mention this in this article since it was carried out after the article was submitted with different participants. But still, it points to the importance of this work. We have found that the dip in conductance near a new channel is due to zeros in the lowest transmission eigenvalue. We find that there are two types of zeros: (1) zeros due to singularities in the complex frequency plane, and (2) zeros that are due to the velocity of the new eigenchannel that enters at the crossover, which is zero at the crossover. For (1) the energy density vanishes at the sample output, while for (2) the energy density is finite at the sample output. We also find that the DOS peaks around the crossover to a new channel. And, in addition, there is a sharp feature in the DOS just at the crossover due to the DOS in the lowest eigenchannel due to the low value of the EV. This appears to be the source of the Wigner cusp anomaly.

We thank Reviewer #1 for encouraging us to write more expansively about the significance of this paper.

Reviewer #2

We followed all the detailed suggestions of the Reviewer #2 and also made other changes along the lines of these comments.

- Lines 12-16: Many of these examples concern incoherent or particle diffusion, in which interference phenomena such as eigenchannels play no role at all. The findings of this study are totally irrelevant to those phenomena.

---The first sentence of the abstract is removed

- Nowhere is a clear definition of the velocity of an eigenchannel to be found. One has to assume that what is meant is the properly weighted average of the group velocity.

---Most generally, the eigenchannel velocity (EV) on the output surface is the average over the angular distribution of light of the longitudinal component of velocity of waves emerging from the sample. The EV on the incident surface is defined similarly for the wave entering the sample. In the quasi-1D geometry the EV is precisely as described by reviewer, as stated, “the properly weighted average of the group velocity.”

---the definition is brought up earlier now in the text, but, in the abstract, we have added the word “longitudinal” that should help.

The definition of eigenchannel velocity is now given the first time this term is used in the Introduction:

“We focus on the longitudinal components of the transmission eigenchannel velocities (EVs), v_n , which are the weighted averages over the angular distribution of the velocity component of the wave normal to the sample surface for different TEs. In the waveguide geometry this may be computed as the weighted averages over the distributions of group velocities of waveguide modes.”

The EVs are expressed mathematically, the next time they are mentioned, at the point where the mathematical definition is first needed.

- L39: The sentence is really unnecessarily hard to read.

---The discussion here is now broken into several shorter sentences:

“The limits of diffusive propagation may also be given in terms of the ensemble average of the dimensional conductance, g , which is equivalent to the average of the classical transmittance, $g = \langle T \rangle$ ^{26,27}. The dimensionless conductance is the average electronic conductance in units of the quantum of conductance, $\frac{e^2}{h}$, while the transmittance is the sum over all pairs of flux transmission coefficients between the N incident and outgoing channels of the sample, $T = \sum_{a,b}^N |t_{ba}|^2 = \langle \text{Tr}(tt^\dagger) \rangle$ ^{10,28,29}. Here, t is the transmission matrix (TM) with elements t_{ba} .”

The effort has been made throughout the text to break down complex sentences.

- L49 in the case of particle diffusion, no speckle pattern occurs. In the case of quantum diffusion there can be a speckle of probability density but not of particle density.

---The point of this discussion is to show that light and electron propagation can be treated on the same footing, so that the work presented here is relevant to the electronic context. Much of this discussion has now been eliminated to keep the text simple, but the point can still be made in a simpler way. Viewing propagation of electrons and photons from a particle perspective gives a similar picture, as does viewing their propagation from a wave perspective. It is just a question in each case of which is called "classical" or "quantum." So, from this perspective, it would be fair to say that there is a stable pattern of the amplitude squared of the electronic wavefunction or the electromagnetic field in a static sample. One could say the optical speckle pattern gives the probability of finding a photon, while the electronic speckle pattern is the probability of finding an electron. Since the optical and electronic cases can be treated on the same footing, following Landau and Landauer, all the finding here for electromagnetic radiation apply equally to quantum transport of electrons. With this in mind, I have changed the text to read.

"The interference of classical and quantum mechanical waves in mesoscopic samples produces a stable speckle pattern of the square amplitude of the electromagnetic and quantum waves, respectively."

- L74 here it is unclear what the τ_n represent. The authors use tau for times and t for transmission coefficients, except sometimes they use t for times and tau for transmission coefficients. Please clarify by clearly defining each symbol, and if possible use the same symbol for the same quantity.

---the question of proper symbols is something we have struggled with and decided in earlier papers. Please allow me to explain. We cannot use τ_n to represent the transmission time in the nth eigenchannel since this symbol is well-established as denoting the nth transmission eigenvalue. The symbol for transmission time, τ_T was chosen to fall in line with the symbol for the Wigner time, τ_W . In many circumstances, it is important to compare and contrast these two symbols.

- L80-81 Ultracold atoms. this line seems out of place.

--The idea here was to expand the applicability of the present work. Just as there are strong similarities between electron and electromagnetic propagation, there are strong similarities with Bose-Einstein condensates. I agree it is best to keep things simple and I have removed this sentence.

- L92 define τ_n

---The dimensionless conductance may be expressed as the sum of the N transmission eigenvalues, $g = \langle \text{Tr}(tt^\dagger) \rangle = \sum_{n=1}^N \tau_n = \langle T \rangle^{10-12,14}$

The τ_n are defined here as the transmission eigenvalues. But to be clear, I have now broken this discussion into several sentences:

"The limits of diffusive propagation may also be given in terms of the ensemble average of the dimensional conductance, g , which is equivalent to the average of the classical transmittance,

$g = \langle T \rangle$ ^{26,27}. The dimensionless conductance is the average electronic conductance in units of the quantum of conductance, $\frac{e^2}{h}$, while the transmittance is the sum over all pairs of flux transmission coefficients between the N incident and outgoing channels of the sample, $T = \sum_{a,b}^N |t_{ba}|^2 = \langle \text{Tr}(tt^\dagger) \rangle$ ^{10,28,29}. Here, t is the transmission matrix (TM) with elements t_{ba} . The TM is most often applied to the quasi-1D wire or waveguide geometry with constant cross section and reflecting sides. A natural choice for the channels is the set of the N propagating modes of the empty waveguide. The eigenvalues of tt^\dagger are the transmission eigenvalues, τ_n , so that $T = \sum_{n=1}^N \tau_n$, with τ_n decreasing for increasing n ."

- L106 v_n is not defined precisely

---We now define v_n in the penultimate sentence of the Introduction, as mentioned above. The full text of the first mention of the v_n is given below:

"In this study, we show that the velocity distributions of TEs on the input and output surfaces of multiply scattering media are not randomized even by multiple scattering. We focus on the longitudinal components of the transmission eigenchannel velocities (EVs), v_n , which are the weighted averages over the angular distribution of the velocity component of the wave normal to the sample surface for different TEs. In the waveguide geometry this may be computed as the weighted averages over the distributions of group velocities of waveguide modes."

Since this is in the Introduction, we tried to keep this discussion as simple as possible. A few paragraphs later in the Results section, we give the mathematical expression for the v_n .

- L135 how are the eigenvalues sorted? High to low or low to high?

---We now make this clear when the τ_n are first mentioned.

"The eigenvalues of tt^\dagger are the transmission eigenvalues, τ_n , so that $T = \sum_{n=1}^N \tau_n$, with τ_n decreasing for increasing n ."

- L176 repetition of the formula?

---The second instance of this formula in the same paragraph is now removed

- L177 This seems to be true only in an ensemble average sense

---This is now made clear by adding the word "average" in the sentence:

"Since the average of ρ_ω is independent of scattering strength in systems with the same average value of ϵ , it is the same as in a homogeneous sample⁵⁸."

- L178 LDOS should be a strongly fluctuating function (unless one means after ensemble averaging)

---the use of the phrase “average of ρ_ω ” in the sentence discussed above now corrects this problem.

- L230 scattering length – do the authors mean scattering mean free path?

---This has now been changed everywhere it occurred both in the main text and in the Supplementary Information. The phrase “scattering time” has all been changed to “scattering mean free time”.

- 248 Unlike, ... The sentence makes no sense

---The sentence has now been replaced with the following:

“Unlike, the Boltzmann diffusion coefficient, D_B , the diffusion coefficient, D , is expressed here in terms of the nature of the wave near the boundaries rather than in the bulk of the medium. As seen below, D and its factors in Eq. (7) depend upon the dimensions of the sample even in the diffusive limit.”

- L302 Falls 10% higher

---The sentence is now:

In addition to the larger magnitude of the slope of $u(z)$ in the centre relative to that at the boundaries of the sample as g decreases, as seen in Fig. 6(a), there is an inversion in the ranking of energy excited in the sample vs. eigenchannel index, n , in the crossover from ballistic to diffusive propagation.

- L368 Multiple scattering

On pages 19 and further the influence on the width of the conductance, and especially the nonmonotonic behavior upon opening a new channel is discussed. This effect has been observed in calculations and numerical simulations, see e.g. <http://iopscience.iop.org/0295-5075/24/4/006> and Garcia-Mochales, P., et al. "Conductance in disordered nanowires: Forward and backscattering." *Physical Review B* 53.15 (1996): 10268.

It is well known that in the so-called semi-ballistic or quasi-ballistic regime, where some of the sample dimensions are not large with respect to the scattering mean free path, a remnant of the resonance at opening of a new channel remains. Since the dimensions of the system in these directions is not much larger than the mean free path, the conditions for applying the scaling theory in these directions are not fulfilled and the discussion on L368-373 makes no sense.

---This is something I missed! I cannot thank the reviewer enough for pointing out this key literature. We now reference this work in Ref 75-78 and discuss the experimental work on conductance and transmittance for ballistic waves, and the simulations of diffusive conductance in the references sent by the reviewer. Our paper did not discuss the transverse scaling of conductance, but the variation of conductance with transverse dimensions is a key precedent to the transverse scaling of the parameters of interest in this paper, D , z_b , v_T , and ℓ_s .

-L395 Please give concrete examples of how the results of the present work relate to specific applications in medical imaging and interventions, telecommunication, and sensing. Or put a less definitive statement.

---I believe there are strong connections of the present work to all these fields but have now refrained from mentioning this in the conclusion. Trying to be specific would open up a subject that is much more actively studied than the fundamental questions explored in the present paper. It would therefore be cleaner not to veer away from the main thesis presented.

I should say that in a previous paper in Nature Communications, presently reference 45, we explored the energy density profile of TEs in the sample in the sample and tried to come up with a theoretical formula for this. There are some small differences between our expressions and the simulations. I now realize this was because we incorrectly assumed, as everyone else in this field had, that the velocities of TEs in multiply scattering media are randomized and that all TEs have the same velocities.

I am greatly indebted to the Reviewer #2 for so many helpful suggestions. Addressing them has greatly improved the paper and saved us from not properly crediting relevant research.

REVIEWERS' COMMENTS

Reviewer #1 (Remarks to the Author):

The paper has been significantly improved taking into account the points that were raised. I think the paper is ready for publication now.